# LLM2Fx-Tools:
# Tool Calling For Music Post-Production

**Seungheon Doh**[1,2,*,†]    **Junghyun Koo**[2,*]    **Marco A. Martínez-Ramírez**[2]    **Woosung Choi**[2]
**Wei-Hsiang Liao**[2]    **Qiyu Wu**[3]    **Juhan Nam**[1]    **Yuki Mitsufuji**[2,3]
[1]KAIST    [2]Sony AI    [3]Sony Group Corporation
seungheondoh@kaist.ac.kr    junghyun.koo@sony.com

## Abstract

This paper introduces LLM2Fx-Tools, a multimodal tool-calling framework that generates executable sequences of audio effects (Fx-chain) for music post-production. LLM2Fx-Tools uses a large language model (LLM) to understand audio inputs, select audio effects types, determine their order, and estimate parameters, guided by chain-of-thought (CoT) planning. We also present LP-Fx, a new instruction-following dataset with structured CoT annotations and tool calls for audio effects modules. Experiments show that LLM2Fx-Tools can infer an Fx-chain and its parameters from pairs of unprocessed and processed audio, enabled by autoregressive sequence modeling, tool calling, and CoT reasoning. We further validate the system in a style transfer setting, where audio effects information is transferred from a reference source and applied to new content. Finally, LLM-as-a-judge evaluation demonstrates that our approach generates appropriate CoT reasoning and responses for music production queries. To our knowledge, this is the first work to apply LLM-based tool calling to audio effects modules, enabling interpretable and controllable music production. [1]

## 1 Introduction

The use of Audio effects (Fx) processing constitutes a fundamental component of modern music post-production, where producers systematically apply sequences of effects (Fx-chain) to transform audio signals and achieve desired sound textures (Zölzer et al., 2002; De Man & Reiss, 2013). In most post-production workflows, the application of audio effects is guided by both technical and creative criteria, requiring a high level of expertise from audio engineers. Estimating the appropriate Fx-chain from unprocessed input audio (dry), processed output audio (wet), or by reverse engineering from both requires extensive domain expertise and often involves iterative manual adjustment.

To address this challenge, automatic Fx-chain estimation has emerged as a promising approach to reduce barrier and labor-intensive aspects of music production. Previous works focus on signal processing-based optimization (Barchiesi & Reiss, 2010; Giannoulis et al., 2013; Ma et al., 2015), gradient-based optimization (Colonel & Reiss, 2021; Lee et al., 2024; Steinmetz et al., 2024; Koo et al., 2025; Lee et al., 2025; Yu et al., 2025), regression methods (Rämö & Välimäki, 2019; Sheng & Fazekas, 2019; Mimilakis et al., 2020; Martínez-Ramírez et al., 2021; Steinmetz et al., 2022; Hayes et al., 2025), and multitask methods (Mitcheltree & Koike, 2020; Lee et al., 2023; Take et al., 2024). While these methods demonstrate promising performance, they face several fundamental limitations. First, gradient-based methods require differentiable audio effects modules, limiting their applicability to specific effects. Second, regression and signal processing-based methods operate on fixed, predefined configurations and lack the ability to dynamically select effects and determine their ordering. Furthermore, these approaches lack user-level interpretability, as they provide only Fx-chain without human-readable descriptions or reasoning to explain why such decisions are made.

Meanwhile, recent advances in large language models (Achiam et al., 2023; Grattafiori et al., 2024; Comanici et al., 2025) (LLMs) have introduced powerful capabilities including instruction follow-

---

[*]Equal contribution
[†]Work done while an intern at Sony AI
[1]Demo is available at: https://seungheondoh.github.io/llm2fx-tools-demo/

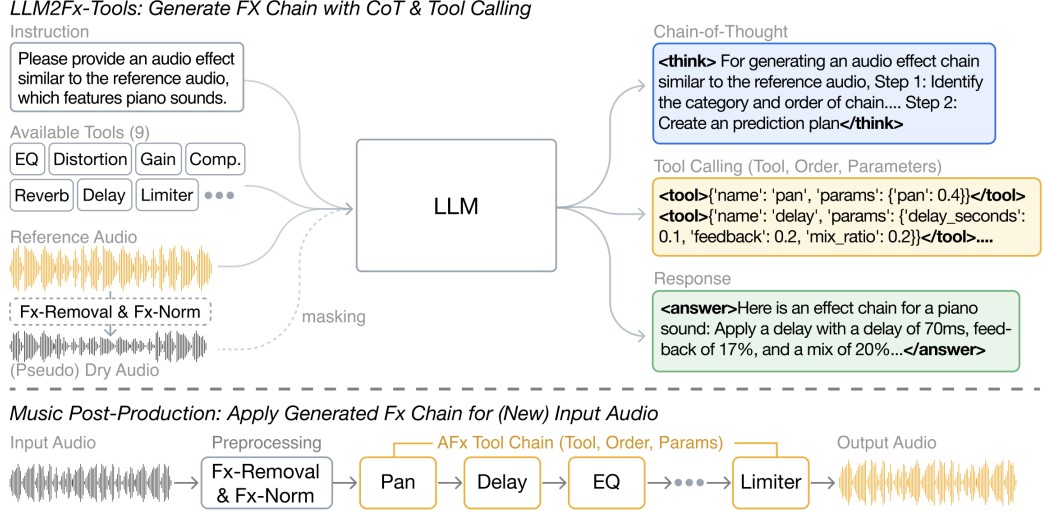

Figure 1: An illustration of the LLM2Fx-Tools framework. The input to LLM2Fx-Tools consists of instruction, available tools, reference audio, and (pseudo) dry audio that is preprocessed with audio effects removal and noramlization (Fx-Removal and Fx-Norm). The framework outputs chain of thought, tool calling procedure, and response. The generated tool calling outputs (Fx-chain) are then combined with tool environments (audio effects modules) to enable the transformation of new audio in the style of the reference audio.

ing (Achiam et al., 2023; Wei et al., 2022a), chain-of-thought reasoning (Wei et al., 2022b), and tool calling (Schick et al., 2023). Chain-of-thought (CoT) enables models to decompose a complex task into a series of reasoning sub-tasks, providing an interpretable view of its reasoning process (Wei et al., 2022b). Tool calling enables LLMs to flexibly connect with external modules (e.g., non-differentiable audio effect modules) and knowledge bases to accomplish domain specific tasks (Schick & Schütze, 2020; Gao et al., 2023; Doh et al., 2025b). These capabilities present new opportunities to address the flexibility and interpretability issues in Fx-chain prediction. In the context of music post-production specifically, LLM2Fx (Doh et al., 2025c) predicts audio effects parameters from natural language prompts, but it does not employ explicit tool calling or chain-of-thought reasoning and is limited to single effects (Equalization and Reverb).

In this work, we introduce LLM2Fx-Tools, LLM-based Fx-chain estimation with Tool-calling, a multimodal framework that addresses these limitations by enabling flexible Fx-chain prediction through tool calling and enhancing interpretability with chain-of-thought reasoning. LLM2Fx-Tools generates 1) chain-of-thought, 2) executable Fx-chain and 3) natural language response. Our key contributions are:

**Tool-Calling for Music Production:** We develop the first structured tool-calling approach for Fx-chain generation that enables multimodal LLMs to understand audio conditioning and generate executable tool calls for non-differentiable audio effects modules.

**Chain-of-Thought for Fx-chain Planning:** We utilize a chain-of-thought (CoT) mechanism designed for Fx-chain generation that decomposes the complex task into interpretable sub-tasks: effect selection, order determination, and parameter estimation. This intermediate reasoning bridges the gap between user inputs and target Fx-chain, improving both performance and interpretability.

**Multimodal Instruction-Following:** We extend the Fx-chain estimation task from unimodal audio-to-effects mapping to a multimodal framework incorporating natural language instructions. Users can specify preferred effect types, musical genres, or instrument characteristics, enabling customized Fx-chains that align with specific user requirements.

**Conversational Music Production Dataset:** We introduce LP-Fx, LLM-based music production dataset for audio effects tools, containing 101K conversational examples with structured Tool Calling, Chain-of-Thought, and Response. Each example comprises 1) user instructions, 2) audio effects tool calls, 3) chain-of-thought, and 4) responses.

## 2 LLM2Fx-Tools: Fx-chain Generation via Tool Calling

### 2.1 Task Definition

Our main task is to estimate the Fx-chain ($\mathcal{C}$) that transforms a dry unprocessed audio ($x_{\text{dry}}$) into a reference audio signal ($x_{\text{ref}}$). Formally, given $x_{\text{dry}}$ and $x_{\text{ref}}$, our goal is to estimate $\mathcal{C}$ such that $x_{\text{ref}} = \mathcal{E}(\mathcal{C}, x_{\text{dry}})$, where $\mathcal{E}$ denotes the tool environment that applies $\mathcal{C}$ to $x_{\text{dry}}$. For additional controllability, we incorporate natural language instructions ($x_{\text{instruction}}$) to guide the generation process. Our goal is to learn the inverse mapping

$$\mathcal{C} = f_\theta(x_{\text{instruction}}, x_{\text{dry}}, x_{\text{ref}}; \mathcal{T}) \tag{1}$$

where $f_\theta$ represents an LLM that predicts the Fx-chain $\mathcal{C} = [(\text{tool}_n, \text{params}_n)]_{n=1}^N$, which consits of a sequence of $N$ tools and their corresponding parameters, from the reference audio $x_{\text{ref}}$ given the available tool set $\mathcal{T}$. We treat each audio effect module $\text{tool} \in \mathcal{T}$ as an external executable tool.

As implied in Eq. (1), we consider Fx-chain estimation task where both $x_{\text{dry}}$ and $x_{\text{ref}}$ are available, which is commonly referred to as the *reverse engineering* (Lee et al., 2025) task. However, $x_{\text{dry}}$ is not always accessible in practical scenarios, corresponding to the *blind estimation* (Lee et al., 2023) task. While our primary target is the reverse engineering, we propose a robust training method to simultaneously handle both *reverse engineering* and *blind estimation* within a single model, as detailed in Section 2.3.

In addition to Fx-chain estimation, our model also aims to generate chain-of-thought ($x_{\text{cot}}$) and natural language responses ($x_{\text{response}}$). The chain-of-thought (CoT) reasoning serves as an intermediate planning stage that decomposes the complex Fx-chain generation into four sequential components: 1) user input analysis, 2) audio effects module selection, 3) processing order determination, and 4) parameter planning. In our auto-regressive generation framework, the CoT functions as an context condition (Wei et al., 2022b) for subsequent tool calling, bridging user queries and action plans to support more accurate and interpretable tool execution. Following the tool calling generation, the model produces natural language responses that provide users with a conversational interface for music production tasks. The overall framework is depicted in Figure 1.

### 2.2 Architecture

To enable LLMs to comprehend audio inputs for tool calling, we adopt a multimodal autoregressive generation framework (Liu et al., 2023; Gardner et al., 2023). As illustrated in Figure 2, since LLMs inherently lack audio processing capabilities, we bridge this modality gap through a pretrained audio encoder coupled with a learnable audio-language adapter. This adapter projects audio representations into the language model's embedding space, formally defined as: $e_{\text{audio}} = f_{\text{adapter}}(f_{\text{encoder}}(x_{\text{audio}}))$, where $f_{\text{encoder}} : \mathbb{R}^{c \times t} \to \mathbb{R}^{l \times d_{\text{enc}}}$ processes input audio signals $x_{\text{audio}}$ ($c$ channels, $t$ samples) to extract $l$ audio representations of dimension $d_{\text{enc}}$, and $f_{\text{adapter}} : \mathbb{R}^{l \times d_{\text{enc}}} \to \mathbb{R}^{l \times d_{\text{LLM}}}$ maps these representations to the language model's embedding space of dimension $d_{\text{LLM}}$. The resulting audio embeddings $e_{\text{audio}}$ are concatenated with text token embeddings to

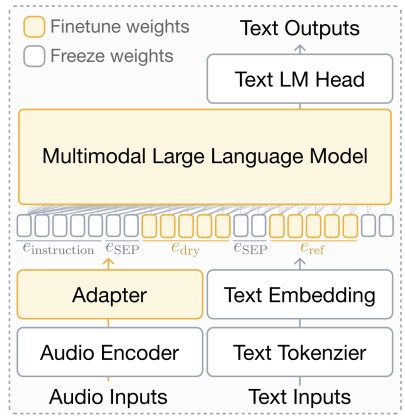

Figure 2: Model Architecture

form a unified multimodal input sequence. Specifically, the input to the LLM is a combined token sequence consisting of the instruction tokens $e_{\text{instruction}}$, separation tokens $e_{\text{SEP}}$ (implemented as natural-language tokens such as "dry audio" and "reference audio" to create token overlap between $e_{\text{instruction}}$ and $e_{\text{audio}}$), and the audio embeddings $e_{\text{dry}}$ and $e_{\text{ref}}$ from the input audio signals $x_{\text{dry}}$ and $x_{\text{ref}}$. This unified multimodal sequence enables the LLM to generate structured outputs including chain-of-thought reasoning $x_{\text{cot}}$, tool calling sequences $\mathcal{C}$, and natural language responses $x_{\text{response}}$.

**Audio Encoder:** We employ Fx-Encoder++ (Yeh et al., 2025), a specialized audio encoder trained with contrastive learning to obtain representations for audio effects processing. This encoder consists of a ConvNet-based feature extractor, pooling layer, and MLP projection layer. We remove the pooling and MLP projection layers and use the patch embeddings $h_{\text{audio}} \in \mathbb{R}^{l \times d_{\text{enc}}}$ as audio representations.

**Adapter:** Unlike previous work that uses a simple linear projection layer for cross-modal alignment (Liu et al., 2023; Gardner et al., 2023), we employ a transformer-based audio-language adapter (Li et al., 2023) with a linear projection layer $W \in \mathbb{R}^{d_{\text{enc}} \times d_{\text{LLM}}}$ and that utilizes 32 learnable query embeddings $e_{\text{query}} \in \mathbb{R}^{32 \times d_{\text{LLM}}}$. This design uses cross-attention to aggregate audio information into learnable query tokens.

**Large Language Model:** We employ Qwen3-4B (Yang et al., 2025) as our foundation LLM backbone, which provides inherent capabilities for structured tool calling and chain-of-thought reasoning. We fine-tune the model using Low-Rank Adaptation (LoRA) (Hu et al., 2022) with rank 128 and alpha 256 to efficiently adapt the model to our Fx-chain estimation task.

## 2.3 TRAINING

We employ a unified autoregressive next-token prediction objective to train our multimodal LLM.

**Cross-Entropy for Next-token Prediction:** Given a training sample with user instruction $x_{\text{instruction}}$, reference audio $x_{\text{ref}}$, dry audio $x_{\text{dry}}$, chain-of-thought $x_{\text{cot}}$, tool calling sequence $\mathcal{C}$, and assistant response $x_{\text{response}}$, we construct the input sequence as a concatenation of a conditioning prefix and a target sequence to be generated by the model as follows:

$$x_{\text{input}} = [\underbrace{x_{\text{instruction}}, x_{\text{dry}}, x_{\text{ref}},}_{\text{Conditioning Prefix}} \underbrace{x_{\text{cot}}, \mathcal{C}, x_{\text{response}}}_{\text{Target Sequence}}] \quad (2)$$

We train the model with the cross-entropy loss $\mathcal{L}_{\text{CE}}$, computed only over the target sequence, while leaving prefix as a conditioning context as follows:

$$\mathcal{L}_{\text{CE}} = -\sum_{t \in T_{\text{target}}} \log p(x_t | x_{<t}; \theta) \quad (3)$$

where $T_{\text{target}}$ represents the set of token indices of the target sequence.

**Number Token Loss:** Standard cross entropy loss treats all incorrect predictions equally, even when some numerical values are closer to the correct answer than others. As a result, directly applying it is not ideal for parameter estimation. To address this limitation, we adopt a regression-like Number Token Loss (NTL), which uses the Wasserstein-1 distance between the predicted and one-hot number distributions (Zausinger et al., 2025):

$$\mathcal{L}_{\text{NTL-WAS}} = \frac{1}{|\mathcal{I}_{\text{num}}|} \sum_{i \in \mathcal{I}_{\text{num}}} \sum_{v \in \mathcal{V}_{\text{num}}} \hat{P}_i(v) |y_i - \text{val}(v)| \quad (4)$$

where $\mathcal{I}_{\text{num}}$ is the set of sequence positions where number-token applies, and $\mathcal{V}_{\text{num}}$ is the subset of tokens corresponding to numeric tokens. For a given position $i$, $\hat{P}_i(v)$ is the predicted probability of token $v$, $y_i$ is the ground truth numerical value, and $\text{val}(v)$ maps token $v$ (strings) to its numerical values (floats). This loss function penalizes predictions based on how far they are from the true numerical value, rather than treating all incorrect tokens equally. Our final loss function combines both objectives: $\mathcal{L}_{\text{total}} = \mathcal{L}_{\text{CE}} + \lambda \mathcal{L}_{\text{NTL}}$ where $\lambda$ is a hyperparameter for balancing cross-entropy and number token losses.

**Multi-Stage Training:** To effectively train our multimodal LLM, we adopt a multi-stage training strategy (Liu et al., 2023) that systematically builds capabilities from basic audio-language alignment to complex reasoning tasks. Our training protocol comprises two distinct phases: 1) modality alignment pre-training and 2) LLM fine-tuning while progressively incorporating task complexity.

We first pre-train the adapter module to bridge the audio modality and text modality. We use only audio inputs and tool calling outputs (Fx-chain) as training data, focusing solely on learning the relationship between dry and reference audio differences and their corresponding Fx-chains. We employ random Fx sampling to maximize the diversity of parameter-audio mappings, ensuring comprehensive coverage of the parameter space and understanding of audio representations. In this stage, we freeze the LLM parameters and only update the audio-language adapter parameters.

In the fine-tuning stage, we initialize the adapter with the pre-trained weights from the previous stage and update both the adapter and LLM through LoRA adaptation (Hu et al., 2022). This stage incorporates the full complexity of our task by training on the complete conversational data, including user instructions, chain-of-thought reasoning, natural language responses, and tool calling sequences.

**Robust Training Techniques for Distribution Shift:** As mentioned in Section 2.1, our goal is to estimate $\mathcal{C}$ such that $x_{\text{ref}} = \mathcal{E}(\mathcal{C}, x_{\text{dry}})$, assuming we have access to both $x_{\text{dry}}$ and $x_{\text{ref}}$. However, training a model only on a dataset of paired ($x_{\text{dry}}$, $x_{\text{ref}}$) audio samples for the *reverse engineering* setup pose distribution shift challenges in real-world scenarios, where $x_{\text{dry}}$ is typically unavailable during inference. Even when it is available, its acoustic environment may differ from the training distribution since recording studios vary significantly in their equipment and environments.

To address this challenge, we preprocess the input audio by employing Fx-Removal (Rice et al., 2023) and Fx-Normalization (Martínez-Ramírez et al., 2022) techniques at both the training and inference stages, in order to align environmental distributions and obtain pseudo-dry audio $\hat{x}_{\text{dry}}$. Furthermore, we apply dry audio masking during training, randomly omitting dry audio inputs with probability $p_{\text{masking}}$ to force the model to rely solely on reference audio for the *blind estimation* setup.

## 3    DATASET: LP-FX

Following previous works (Doh et al., 2025a; Choi et al., 2025), we adapt the LLM based data synthesis pipeline and further improve it by incorporating an LLM-as-a-judge (Zheng et al., 2023), enabling the systematic generation of high-quality conversational data for Fx-chain generation tasks.

### 3.1    BASE DATASET AND TOOL ENVIRONMENT

The audio source of LP-Fx is MedleyDB (Bittner et al., 2014; 2016), which provides royalty-free 196 multitrack recordings. Each recording includes three different levels of audio: (i) unprocessed *raw* tracks, (ii) *stems*, which are submixes of raw tracks with audio effects applied, and (iii) a full *mix*, created by combining the processed stems into a complete mixture. We use unprocessed raw audio as the dry audio $x_{\text{dry}}$. We filter out multitracks with bleed using the metadata provided by MedleyDB[2], resulting in a curated set of 2,119 raw audio files from 116 multitracks, spanning 9 genres and 80 unique instruments. We use the `Pedalboard`[3] audio effects library and our custom audio effects modules as our tool environment $\mathcal{T}$. We select 6 modules (compressor, distortion, reverb, delay, limiter, and gain) from the Pedalboard library and 3 modules (three-band equalizer, stereo widener, and panner) from our custom modules, totaling 9 modules and 26 parameters.

### 3.2    DATA GENERATION PROCESS

As illustrated in Figure 3, our data generation process consists of three sequential stages. In the first stage, we sample Fx-chains within musically plausible ranges to create dry/processed audio pairs. The second stage generates instruction-following conversations grounded in these Fx-chains to ensure factual accuracy. The third stage produces chain-of-thought reasoning that explicitly connects user instructions to the underlying audio effects transformations. Finally, we employ LLM-as-a-judge (Chen et al., 2024; Zheng et al., 2023) evaluation to filter the dataset for high-quality samples. The basic framework for utilizing LLMs in our data generation process follows

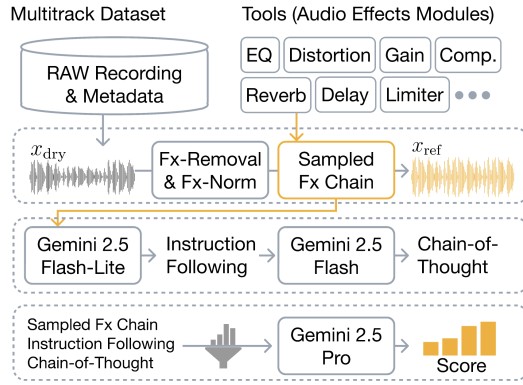

Figure 3: Data generation process for LP-Fx

the formulation $x_{\text{data}} = \text{LLM}(x_{\text{ground}}, \mathcal{P})$, where $x_{\text{ground}}$ is the grounded information (e.g., Fx-chain $\mathcal{C}$) and $\mathcal{P}$ is the carefully crafted prompts. We further provide details of each stage in Appendix C.

### 3.3    STATISTICS

Table 1 presents the statistics of our LP-Fx dataset. Based on the observation that task complexity increases with the number of effects in the chain, we create a stratified dataset with 11,100 training

[2]https://github.com/marl/medleydb/blob/master/ERRATA.md
[3]https://github.com/spotify/pedalboard

samples and 100 test samples for each Fx-chain length from 1 to 9, resulting in a total of 100,800 conversations. We ensure no overlap between audio tracks used in training and test sets to prevent data leakage.

Each example comprises 1) user instructions, 2) unprocessed/processed audio pairs, 3) executable audio effects tool calls, 4) chain-of-thought reasoning, and 5) assistant responses. The dataset's rich diversity in musical content makes it particularly effective for LLM fine-tuning. With 2,019 tracks spanning 80 unique instruments across 9 genres.

Table 1: Statistics of LP-Fx Dataset.

| Statistics | Trainset | Testset |
|---|---|---|
| # of Dialogue | 99,900 | 900 |
| # of Tracks | 2019 | 100 |
| # of Instruments | 80 | 33 |
| Avg. Instruction length | 28.8 | 28.2 |
| Avg. Response length | 178.6 | 178,6 |
| Avg. CoT length | 251.9 | 252.7 |
| Min/Max Tool Number | 1-9 | 1-9 |

## 4 EXPERIMENTS

### 4.1 REVERSE ENGINEERING

**Task Definition:** We evaluate our approach on *reverse engineering*. The task involves predicting Fx-chains $\mathcal{C}$ from reference audio $x_{ref}$ given access to the corresponding dry audio $x_{dry}$. We evaluate this task using the LP-Fx test split, which provides ground truth triplets of $(x_{dry}, x_{ref}, \mathcal{C})$ for evaluation.

**Metrics:** Our evaluation framework assesses model performance through four complementary perspectives: 1) *Fx-chain Planning*, 2) *Perceptual Distance*, 3) *DSP Feature Distance*, and 4) *Deep Embedding Similarity*. For *Fx-chain Planning*, we use classification accuracy to measure whether the model correctly predicts the presence of target audio effects modules in the ground truth Fx-chain. We then apply Spearman rank correlation to assess how closely the predicted ordering of modules matches the ground truth. Finally, parameter mean absolute error (MAE) is used to quantify the precision of parameter prediction. For *Perceptual Distance*, we employ Multi-Resolution STFT (MRS) distance (Yamamoto et al., 2020) on both left-right (L/R) and mid-side (M/S) channels for stereo-aware processing evaluation. For *DSP Feature Distance*, we utilize Audio Features (AF) (Man et al., 2014; Vanka et al., 2024), including root mean square, crest factor, stereo width, stereo imbalance, and bark spectrum. For *Deep Embedding Similarity*, we employ audio effects-specific pretrained representations, including classification-based AFx-Rep (Steinmetz et al., 2024) and contrastive learning-based Fx-Encoder (Koo et al., 2023).

**Baselines:** We evaluate our approach against several baselines to assess the effectiveness of our instruction-following Fx-chain generation framework.

*1) No Fx:* A naive baseline that applies no audio effects to the input audio, representing the lower bound performance where the predicted reference audio $\hat{x}_{ref}$ equals the dry audio $x_{dry}$.

*2) Random Fx:* The random number of effects with randomized ordering and parameters.

*3) Regression:* A regression-based approach that directly predicts audio effects parameters from audio features without explicit tool selection or ordering capabilities. Specifically, we first extract embeddings using the Fx-Encoder++ (Yeh et al., 2025), followed by a 2-layer MLP with ReLU activations. The regression head outputs a vector of logits corresponding to the number of parameters in the full Fx-chain.

*4) Multitask:* An enhanced regression model incorporating additional classification heads to address limitations of the pure regression approach. The base architecture is identical to the regression model, but additional logits are predicted to classify which audio effects modules are applied.

*5) DeepAFx-ST* (Steinmetz et al., 2022): This approach employs differentiable audio effects modules that enable backpropagation through signal-level objectives, specifically trained with Multi-Resolution STFT loss (Yamamoto et al., 2020). We include this method as a baseline to assess how close its outputs can be in terms of perceptual distance. In our experiments, we extend the original implementation by adapting the differentiable audio effects modules to match the audio effects types used in our system, leveraging the `dasp-pytorch` repository[4].

*6) Gemini 2.5 Flash* (Comanici et al., 2025): A closed-source multimodal LLM with audio understanding, reasoning, and tool calling capabilities.

---

[4] https://github.com/csteinmetz1/dasp-pytorch

### 4.2 AUDIO EFFECTS STYLE TRANSFER WITH BLIND ESTIMATION

**Task Definition:** We evaluate our approach on the audio effects style transfer task, which simulates real-world scenarios where users only have access to reference audio with different content. This task comprises two sequential stages: 1) *blind estimation* - inferring the underlying Fx-chain $\mathcal{C}$ from processed reference audio $x_{\text{ref}}$. Unlike traditional blind estimation approaches, our method provides additional context through Fx Removal and Fx Normalization applied to $x_{\text{ref}}$, enabling reverse engineering-style inference. 2) *style transfer* - applying the estimated Fx-chain to another source audio $x_{\text{input}}$. This evaluation assesses the model's ability to generalize to unseen audio content.

**Evaluation Protocol:** To evaluate generalization across different musical content, we employ MoisesDB (Pereira et al., 2023) as the source of processed reference stems and MedleyDB (Bittner et al., 2014; 2016) as the source of clean input audio. This cross-dataset evaluation protocol ensures that models encounter entirely unseen audio content, providing a test of generalization capabilities. We construct evaluation pairs by matching instrument categories between the two datasets, resulting in an evaluation set of 100 test samples. Given that the reference and input audio contain distinct musical content, we focus our evaluation on feature-based metrics, including DSP feature distance and embedding similarity. We employ the same baseline methods of the *reverse engineering task* as described in 4.1.

### 4.3 NATURAL LANGUAGE GENERATION

**Task Definition:** Beyond the Fx-chain estimation capabilities, LLM2Fx-Tools generates chain-of-thought reasoning and natural language responses, providing interpretability and transparency to users through comprehensive explanations of the audio processing decisions.

**Evaluation Protocol:** We evaluate the natural language generation quality of our LLM2Fx-Tools framework. Following previous works (Gardner et al., 2023; Clemens & Marasović, 2025), we assess the natural language generation capabilities through an LLM-as-a-judge framework (Zheng et al., 2023). We use GPT-5 (OpenAI, 2025) as $\text{LLM}_{\text{judge}}$. Specifically, we evaluate three key dimensions: 1) *tool calling success*, whether the model correctly executes the required Fx-chain, 2) *instruction following quality*, whether the generated response adequately addresses the user instruction, and 3) *chain-of-thought quality*, whether the reasoning effectively connects user instructions to responses through coherent intermediate steps. This process can be formulated as $(s_{\text{IF}}, s_{\text{CoT}}) = \text{LLM}_{\text{judge}}(x_{\text{instruction}}, x_{\text{response}}, x_{\text{cot}}, \mathcal{P}_{judge})$. $\mathcal{P}_{judge}$ details provided in Appendix F.

**Baselines:** We compare our approach with LLMs for natural language generation: *1) Qwen2.5-Omni 7B:* (Chu et al., 2024) An open-source multimodal LLM without reasoning capabilities, *2) Qwen 2.5 4B:* (Yang et al., 2025) A compact open-source LLM without audio understanding, and *3) Gemini 2.5 Flash:* (Comanici et al., 2025) A closed-source multimodal LLM with advanced reasoning capabilities.

### 4.4 TRAINING / EVALUATION DETAILS

We utilize Qwen3-4B (Yang et al., 2025) as our pretrained LLM foundation, which provides instruction following, reasoning and tool calling ability. Training is performed across multi-stage training (MST) with different learning rates and batch sizes. For Stage 1 (modality alignment pretraining), we use a learning rate of 1e-4 with batch size of 32 and train for 100K steps. Stage 2 (LLM finetuning) employs a learning rate of 5e-5 with batch size of 16 and is iterated for 400K steps.

## 5 RESULTS / ANALYSIS

### 5.1 REVERSE ENGINEERING

**Comparison on Fx-chain Planning.** Table 2 demonstrates that LLM2Fx-Tools achieves superior performance across multiple evaluation dimensions. In *Fx-chain Planning*, our approach significantly outperforms all baselines, achieving 80% accuracy in audio effects module classification and 0.56 Spearman correlation for ordering, compared to the MultiTask baseline with 61% accuracy and near-zero correlation. While the regression baseline achieves slightly better parameter MAE (0.20

Table 2: Fx-chain Estimation Results. We compare with multiple baselines and analyze the contribution of key components in our LLM2Fx-Tools framework: Chain-of-Thought (CoT), Number Token Loss (NTL), and Multi-Stage Training (MST). *DeepAFx-ST was trained with Perceptual Dist. as its training objective.

| | Fx-chain Planning | | | Perceptual Dist. | | DSP | Embedding Sim.($\uparrow$) | |
| --- | --- | --- | --- | --- | --- | --- | --- | --- |
| | Acc.($\uparrow$) | Corr.($\uparrow$) | MAE($\downarrow$) | L/R($\downarrow$) | M/S($\downarrow$) | AF($\downarrow$) | AFx-Rep | FxEnc |
| No Fx | - | - | - | 13.11 | 13.49 | 14.82 | 0.50 | 0.30 |
| Random Fx | 52% | -0.01 | 0.39 | 8.07 | 8.90 | 13.70 | 0.41 | 0.34 |
| Regression | 55% | -0.03 | **0.20** | 3.81 | 4.12 | 9.20 | 0.62 | 0.64 |
| MultiTask | 61% | 0.00 | 0.23 | 3.17 | 3.39 | 8.39 | 0.63 | 0.66 |
| DeepAFx-ST | - | - | - | **1.75***| **2.06*** | **3.95** | 0.62 | 0.66 |
| Gemini2.5$_{Flash}$ | 78% | 0.54 | 0.32 | 3.42 | 4.24 | 14.97 | 0.56 | 0.50 |
| LLM2Fx-Tools | **80%** | **0.56** | 0.23 | 3.13 | 3.27 | 8.29 | **0.68** | **0.67** |
| w/o CoT | 67% | 0.49 | 0.24 | 3.34 | 3.38 | 8.39 | 0.64 | 0.66 |
| w/o NTL | 73% | 0.51 | 0.32 | 3.69 | 3.52 | 8.47 | 0.61 | 0.63 |
| w/o MST | 76% | 0.55 | 0.25 | 3.21 | 3.32 | 8.30 | 0.67 | 0.64 |

vs 0.23), this comes at the cost of substantially worse audio effects module selection and ordering capabilities. As DeepAFx-ST is trained directly with an audio domain objective, it exhibits strong performance on audio distance metrics; yet, it lacks the capability to utilize non-differeneitable modules. Interestingly, Gemini 2.5 Flash demonstrates strong *Fx-chain Planning* capabilities with 78% effect classification accuracy and reasonable ordering correlation (0.54). However, it exhibits limitations in parameter estimation, achieving the highest parameter MAE (0.32).

The performance improvements of LLM2Fx-Tools stem from two key aspects: 1) our instruction-following capabilities leverage natural language understanding to provide additional conditioning beyond pure audio comprehension, enabling more precise and semantically-aware audio processing decisions; and 2) the autoregressive sequence modeling inherent in LLMs provides a fundamental advantage for handling Fx-chain ordering compared to models that rely solely on audio features.

**Does Fx-chain Planning Lead to Better Acoustic Similarity?** For *Perceptual Distance*, LLM2Fx-Tools achieves the competitive performance on both MRS distances. Our analysis indicates that effective *Fx-chain Planning* is essential not only for accurate parameter prediction but also for achieving strong perceptual performance. Notably, while the regression baseline achieves the lowest parameter MAE (0.20), this advantage in parameter space does not translate into improved perceptual distance. Because the regression model lacks the ability to selectively apply audio effects modules, it must predict parameters for all predefined modules, even when they are absent in the reference audio. This limitation leads to suboptimal perceptual distance, underscoring the importance of Fx-chain planning for bridging parameter accuracy and perceptual quality.

In contrast, both the MultiTask baseline and our LLM2Fx-Tools framework, which incorporate audio effects module selection capabilities, demonstrate superior performance in both perceptual and DSP distance compared to the base regression approach. Comparing MultiTask vs LLM2Fx-Tools further demonstrates the critical importance of Fx-chain ordering: despite achieving similar DSP distances (8.39 vs 8.29), LLM2Fx-Tools's substantial improvement in ordering correlation (0.56 vs 0.00) leads to better perceptual reconstruction (3.13 vs 3.17 L/R MRS). This indicates that correct effect sequencing significantly contributes to audio processing quality, as the order of effects can dramatically alter the final audio output. For *Deep Embedding Similarity*, LLM2Fx-Tools achieves the highest similarity scores (AFx-Rep: 0.68, Fx-Encoder: 0.67), demonstrating that effective Fx-chain outputs more semantically similar to reference audio.

**Ablation Studies.** The lower portion of Table 2 demonstrates that our core design choices contribute meaningfully to model performance. Chain-of-Thought (CoT) reasoning significantly aids Fx-chain Planning capabilities, improving effect classification accuracy from 67% to 80% and enhancing ordering correlation from 0.49 to 0.56. Number Token Loss (NTL) notably impacts parameter estimation, reducing MAE from 0.32 to 0.23, while also improving overall perceptual and feature-level metrics. MST provides improvements across all metrics, bridging the representations between the pretrained audio encoder and LLM while leveraging the pretrained capabilities of Qwen3.

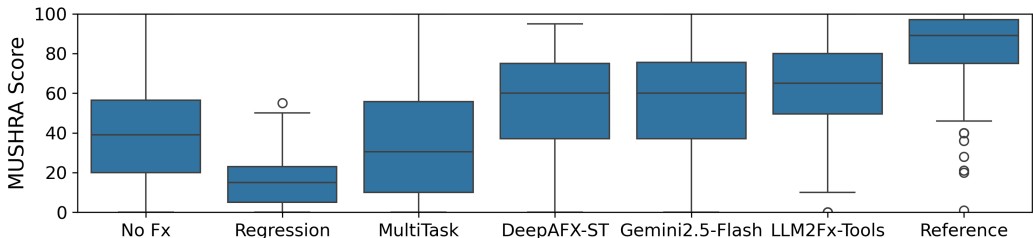

Figure 4: Subjective evaluation on reverse engineering.

**Human Listening Test.** To evaluate the perceptual quality of reverse engineering results, we conduct a MUSHRA (MUltiple Stimuli with Hidden Reference and Anchor) listening test (Series, 2014). Participants were asked to rate different mixes based on their similarity to the reference mix in terms of audio effects characteristics. Each test question consists of a reference track and seven different stimuli tracks. We designed the test with a total of 8 questions and 16 participants.

Figure 4 presents the MUSHRA test results. The hidden reference achieves a mean score of 81.5, confirming the validity of our test design. Our proposed LLM2Fx-Tools outperforms other methods. Pairwise t-tests reveal that LLM2Fx-Tools (62.8), Gemini 2.5 Flash (56.5), and DeepAFX-ST (54.8) all significantly outperform the No Fx baseline (39.1) with $p < 0.05$. While Gemini 2.5 Flash and DeepAFX-ST show no statistically significant difference between each other ($p = 0.747$), LLM2Fx-Tools significantly outperforms both Gemini 2.5 Flash ($p = 0.020$) and DeepAFX-ST ($p = 0.006$). Interestingly, both MultiTask (34.9) and Regression (16.2) baselines score lower than the No Fx baseline, indicating that incorrect effect application can degrade perceptual quality more than applying no effects at all. This finding reinforces our earlier observation that effective Fx-chain planning, including proper effect selection and ordering, is crucial for achieving perceptually similar audio processing results.

## 5.2 AUDIO EFFECTS STYLE TRANSFER WITH BLIND ESTIMATION

Table 3 presents the experiment results of audio effects style transfer, designed to evaluate the cross-domain generalization capabilities of each method. We observe similar trends to those seen in reverse engineering experiments. The regression baseline, which applies all predefined audio effects regardless of their relevance, achieves suboptimal performance with higher AF distance (7.83) and lower embedding similarity scores. The MultiTask approach shows modest improvements with better AF distance (7.62) and enhanced embedding similarities, highlighting the importance of selective effect application. However, its lack of ordering capabilities limits further performance gains. DeepAFx-ST exhibits worse AF distance in the style transfer experiments, indicating limited generalization in this setting.

Among LLM-based approaches, Gemini 2.5 Flash performs poorly, yielding a DSP distance of 9.00 and embedding similarities barely above the No Fx baseline, despite its large parameter count. Our reverse engineering experiments reveal that its parameter predictions are nearly random, which explains why it fails to generalize effectively to the style transfer task. In contrast, LLM2Fx-Tools consistently achieves the best results, with the lowest DSP distance (7.41) and the highest embedding similarity scores (AFx-Rep: 0.35, Fx-Encoder: 0.49) across all evaluated methods.

Table 3: Audio Effects Style Transfer Results.

|  | DSP | Embedding Sim. | |
| --- | --- | --- | --- |
|  | AF($\downarrow$) | AFx-Rep($\uparrow$) | FxEnc($\uparrow$) |
| No Fx | 8.69 | 0.24 | 0.43 |
| Random Fx | 15.22 | 0.14 | 0.19 |
| Regression | 7.83 | 0.24 | 0.31 |
| MultiTask | 7.62 | 0.29 | 0.46 |
| DeepAFx-ST | 10.50 | 0.26 | **0.49** |
| Gemini2.5$_{Flash}$ | 9.00 | 0.24 | 0.27 |
| LLM2Fx-Tools | **7.41** | **0.35** | **0.49** |

## 5.3 NATURAL LANGUAGE GENERATION

Table 4 evaluates natural language generation capabilities through comprehensive LLM-as-a-Judge assessment. Qwen 2.5$_{Omni}$ demonstrates limited zero-shot tool calling capabilities, failing to gener-

Table 4: Natural Language Generation Results. We compare with multiple (multimodal) large language models on tool calling (TC), instruction following (IF), and chain of thought (CoT).

| | Params | Multimodal | Reasoning | TC Success | IF Quality | CoT Quality |
|---|---|---|---|---|---|---|
| Qwen 2.5$_{Omni}$ | 7B | ✓ | ✗ | 0.2% | 1.46 | N/A |
| Qwen 3 | 4B | ✗ | ✓ | 73.7% | 2.89 | 2.30 |
| Gemini 2.5$_{Flash}$ | N/A | ✓ | ✓ | **100%** | 3.39 | 3.03 |
| LLM2Fx-Tools | 4B | ✓ | ✓ | 99.8% | **3.50** | **3.05** |

ate correctly formatted JSON structures. In contrast, Qwen 3-4B achieves substantial tool calling success (73.7%) despite lacking multimodal capabilities, indicating the effectiveness of text-based reasoning for this task. LLM2Fx-Tools, built upon Qwen 3-4B with specialized multimodal training, achieves near-perfect tool calling performance (99.8%), matching the capabilities of state-of-the-art closed-source models such as Gemini 2.5 Flash (100.0%).

The instruction following evaluation reveals substantial quality improvements from domain-specific training. LLM2Fx-Tools outperforms zero-shot baselines, with quality scores increasing from 3.63 to 3.86 compared to the base Qwen 3 model. This improvement demonstrates the critical importance of specialized training for music production tasks, where general-purpose models lack domain-specific knowledge about audio effects and their applications.

Additionally, CoT quality evaluation shows that LLM2Fx-Tools outperforms other LLM baselines. This can be attributed to LLM2Fx-Tools's training on high-quality CoT dataset specifically focused on music production tasks. The training dataset LP-Fx leverages knowledge distillation from Gemini2.5 Flash, with quality assurance provided through filtering by Gemini2.5 Pro. This multi-stage data synthesis approach is expected to enhance the CoT capabilities of the fine-tuned model, enabling more coherent and domain-relevant reasoning for audio effects manipulation.

## 6 LIMITATION

While our framework advances interpretable and controllable Fx-chain estimation, several challenges remain. First, the predicted Fx-chain is interpretable only relative to pseudo-dry audio obtained through Fx-removal and Fx-normalization preprocessing. Full interpretability would require integrating these preprocessing steps directly into the Fx-chain representation and reasoning process. Second, the inherent one-to-many mapping in audio effects estimation (Hayes et al., 2025) creates ambiguity where multiple distinct Fx-chains can produce perceptually similar results, particularly in symmetric parameter spaces. Our current evaluation framework does not fully account for this fundamental ambiguity, and the limited scale of LP-Fx's evaluation set further constrains the ability to comprehensively assess these challenges. Third, our experimental validation focuses exclusively on single-instrument sources, which may limit direct applicability to complex multitrack music production scenarios. Furthermore, we have not evaluated generalization to unseen audio effects modules beyond our training distribution. While our tool-calling framework is designed to be extensible to new VST plugins and audio effects modules, empirical validation of this capability remains future work.

## 7 CONCLUSION

We present LLM2Fx-Tools, a multimodal tool-calling framework for generating executable audio effect chains with interpretable chain-of-thought planning. Our experimental results demonstrate that LLM2Fx-Tools outperforms regression and multitask baselines on both reverse engineering and audio effects style transfer tasks. Additional evaluations through LLM-as-a-judge confirm the natural language generation capabilities of our approach, demonstrating strong instruction following and chain-of-thought reasoning quality. By emitting structured tool calls over audio effect modules, our approach enables controllable and explainable music post-production applications. This work opens several promising directions for future research, including scaling to richer VST plugin toolboxes, developing reinforcement learning frameworks, and conducting comprehensive perceptual evaluations with expert music producers to validate real-world applicability.

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

## A    QUALITATIVE EXAMPLES

In this section, we present a qualitative example below illustrating how LLM2Fx-Tools processes a user query and generates an appropriate Fx-chain. The example highlights each stage of the system's reasoning and output: the user's request, the model's chain-of-thought for Fx-chain modeling and parameter estimation, the corresponding tool calls, and the final response presented to the user.

---

*User Query*
I'm working on a track with Electronic/Fusion synthesizer sounds, and I'd like to emulate
the audio effects style of a reference audio. Could you suggest a plugin chain with a
stereo widener, distortion, and a compressor?

---

*Chain-of-Thought*
For emulating the audio effects style of a reference audio for Electronic/Fusion synthesizer sounds
using a stereo widener, distortion, and compressor:
Step 1. The identified audio effects in the chain are stereo_widener, distortion, and compressor,
in that specific order.
Step 2. For stereo_widener, the width parameter will be adjusted to create an expansive
and immersive stereo image, enhancing the synth's presence.
For distortion, the drive_db will be increased to introduce harmonic richness and saturation,
adding grit and character suitable for electronic fusion.
For compressor, the threshold_db will be set to effectively manage the dynamic range of the synth,
the ratio will be chosen to provide controlled dynamic reduction, the attack_ms will be adjusted to
either preserve or shape the initial transients, and the release_ms will be set to allow the compressor
to disengage smoothly with the synth's decay, ensuring a natural sound.

---

*Tool Calling*
"<tool_call>\n{'name': 'stereo_widener', 'arguments': {'width': 1.3}}\n</tool_call>
<tool_call>\n{'name': 'distortion', 'arguments': {'drive_db': 5.0}}\n</tool_call>
<tool_call>\n{'name': 'compressor', 'arguments': {'threshold_db': -19.0, 'ratio': 4.0,
'attack_ms': 250.0, 'release_ms': 250.0}}\n</tool_call>

---

*Response*
Here is a plugin chain designed to emulate the audio effects style of your reference audio,
featuring a stereo widener, distortion, and a compressor. The stereo widener is set to
a width of 1.3 to expand the stereo image. Following this, a distortion effect with
a drive of 5.0 dB is applied to add harmonic content.
Finally, a compressor with a threshold of -19.0 dB, a ratio of 4.0, an attack of 250.0 ms,
and a release of 250.0 ms is used for dynamic control.",

---

## B    DETAILED TASK DEFINITION

Borrowing notation from (Rice et al., 2023), we define $\mathcal{T}$ as the set of audio effects functions supported by the environment ($\mathcal{E}$). Given a dry audio ($x_{\text{dry}}$), a typical audio processing can be formulated as a composition of functions that yields the processed audio ($x_{\text{wet}}$) as follows:

$$x_{\text{wet}} = g_N \Big( g_{N-1} \Big( \cdots g_2 \big( g_1(x_{\text{dry}}; \phi_1); \phi_2 \big) \cdots ; \phi_{N-1} \Big); \phi_N \Big), \tag{5}$$

where $g_n \in \mathcal{T}$ is the $n$-th effect function and $\phi_n$ denotes the parameters of $g_n$. In this paper, we assume $g_i \neq g_j$ for any pair of $i \neq j$ for simplicity. We denote the Fx-chain by $\mathcal{C} = [(g_n, \phi_n)]_{n=1}^N$ $= [(\text{tool}_n, \text{params}_n)]_{n=1}^N$. Equation 5 can be reformulated in terms of $\mathcal{E}$ and $\mathcal{C}$ as $x_{wet} = \mathcal{E}(\mathcal{C}, x_{dry})$.

Our main task is to reverse-engineer the Fx-chain ($\mathcal{C}$) applied to a reference audio signal ($x_{\text{ref}}$). Specifically, given a processed reference audio signal $x_{\text{ref}}$, we aim to predict the sequence of audio effects and their parameters that were used to create the processed version from an original dry signal ($x_{\text{dry}}$). Formally, we can express this relationship as $x_{\text{ref}} = \mathcal{E}(\mathcal{C}, x_{\text{dry}})$, where the environment ($\mathcal{E}$) applies the Fx-chain ($\mathcal{C}$) to the dry audio ($x_{\text{dry}}$) to produce the processed reference audio ($x_{\text{ref}}$). For additional controllability, we incorporate natural language instructions ($x_{\text{instruction}}$) to guide the generation process. Our goal is to learn the inverse mapping

$$\hat{\mathcal{C}} = f_\theta\Big(x_{\text{instruction}}, x_{\text{dry}}, x_{\text{ref}}; \mathcal{T}\Big), \tag{6}$$

where $f_\theta$ represents an LLM that aims to estimate the original Fx-chain $\mathcal{C} = [(\text{tool}_n, \text{params}_n)]_{n=1}^{N}$ from the reference audio $x_{\text{ref}}$ and $x_{\text{dry}}$ with an additional input $x_{\text{instruction}}$ in the provided environment $\mathcal{T}$. Unlike conventional music captioning tasks (**?**Gardner et al., 2023) that describe the content or characteristics of a single audio signal, our task focuses on identifying the *differences* between two audio signals (Komatsu et al., 2024; Deshmukh et al., 2025), the transformations applied to convert dry audio into processed reference

Our secondary task involves generating intermediate chain-of-thought ($x_{\text{cot}}$) and natural language responses ($x_{\text{response}}$). The chain-of-thought reasoning serves as an intermediate planning stage that decomposes the complex Fx-chain generation into four sequential components: 1) user input analysis, 2) effect selection, 3) processing order determination, and 4) parameter planning. In our autoregressive generation framework, the chain-of-thought functions as an in-context condition for subsequent tool calling, bridging user queries and action plans to support more accurate and interpretable tool execution. Following the tool calling generation, the model produces natural language responses that provide users with a conversational interface for music production tasks.

## C  DETAILED DATASET GENERATION

We detail out each stage of the data generation pipeline for creating LP-Fx below as mentioned in Section 3.2.

**Stage 1: Dry/processed audio pairs synthesis.** For synthesizing processed reference audio $x_{\text{ref}}$, we apply the sampled Fx-chain $\mathcal{C}$ to the dry audio $x_{\text{dry}}$ from MedleyDB. First, we respectively apply Fx-Removal (Rice et al., 2023) and Fx-Normalization (Martínez-Ramírez et al., 2022) (in the order of EQ, stereo imager, and loudness) to the dry audio samples and create a pseudo-dry audio $\hat{x}_{\text{dry}}$. We randomly sample parameters within predefined min-max ranges and quantize them to discrete steps that mirror practical knob granularity (Pestana, 2013). We employ two sampling regimes: a coarse regime to broadly cover the operating space and a fine regime, which reflect real world production practices (De Man, 2017) (sampling ranges are detailed in Table 6). Consequently, we obtain $(\hat{x}_{\text{dry}}, x_{\text{ref}}, \mathcal{C})$ triplets where each triplet contains the original dry audio, the processed reference audio, and the corresponding Fx-chain sequence.

**Stage 2: Instruction-following synthesis.** We synthesize natural single-turn conversations between users and assistants for music production scenarios using the Fx-chains generated in Stage 1. For efficient large-scale generation, we employ a distillation LLM, Gemini-2.5-Flash-lite (Comanici et al., 2025). In this stage, the Fx-chain sequence $\mathcal{C} = [(\text{tool}_n, \text{params}_n)]$ from Stage 1 is paired with task prompts $\mathcal{P}_{chat}$ that describe realistic music production scenarios. The LLM then generates natural language instructions $x_{\text{instruction}}$ and assistant responses $x_{\text{responses}}$ that preserve the underlying Fx-chain structure while providing contextually appropriate explanations, formally expressed as $x_{\text{instruction}}, x_{\text{response}} = \text{LLM}(x_{\text{tool}}, \mathcal{P}_{chat})$.

**Stage 3: Chain-of-thought generation.** To bridge the gap between the Fx-chains $\mathcal{C}$ generated in Stage 1 and the instruction-response pairs $x_{\text{instruction}}, x_{\text{response}}$ from Stage 2, we decompose the music production task into a step-by-step manner. We construct chain-of-thought reasoning by dividing the tool calling process into four sequential steps: 1) user input analysis, 2) tool selection, 3) ordering, and 4) parameter planning. We utilize Gemini-2.5-Flash (Comanici et al., 2025) with enhanced reasoning capabilities for this stage. This process can be formulated as $x_{\text{cot}} = \text{LLM}(x_{\text{instruction}}, x_{\text{response}}, \mathcal{C}, \mathcal{P}_{cot})$ where $\mathcal{P}_{cot}$ represents the task prompts that guide the decomposition of complex audio processing into interpretable reasoning steps.

**Stage 4: Quality filtering.** To ensure data quality and minimize hallucinations in our synthetic dataset, we employ an LLM-as-a-judge evaluation framework using Gemini 2.5 Pro (Comanici et al., 2025). We implement a two dimensional quality assessment evaluating: 1) *tool alignment*, whether generated conversations accurately align with the grounded tool information, Fx-chain $\mathcal{C}$, and 2) *CoT quality*, whether chain of thought reasoning effectively guides

from user queries to tool calling sequences. This process can be formulated as $(s_{\text{tool}}, s_{\text{CoT}}) = \text{LLM}_{\text{judge}}(x_{\text{instruction}}, x_{\text{response}}, x_{\text{cot}}, \mathcal{C}, \mathcal{P}_{judge})$ where $s_{\text{tool}}$ and $s_{\text{CoT}}$ are quality scores of *tool alignment* and *CoT*, respectively, and $\mathcal{P}_{judge}$ is the evaluation prompts. Both $s_{\text{tool}}$ and $s_{\text{CoT}}$ are evaluated using a 4 point Likert scale (from 1=poor to 4=excellent), where samples scoring $\leq 2$ are flagged for regeneration to maintain dataset integrity. $\mathcal{P}_{judge}$ details provided in Appendix F.

## D    ABLATION STUDY ON AUDIO ENCODERS

Table 5 shows an ablation study on the reverse engineering task following the regression approach, as described in Section. 4.1, using different audio encoders. We compare contrastive learning-based music representations (Spijkervet & Burgoyne, 2021; Manco et al., 2022; Doh et al., 2023; 2024; Wu et al., 2025), including audio effects-specific representations (Koo et al., 2023; Steinmetz et al., 2024; Yeh et al., 2025) and audio-text representations (Wu* et al., 2023). We observe that audio effects-specific encoders outperform audio-text representation. With prior work identifying Fx-Encoder++ as the strongest audio-effects representation (Yeh et al., 2025), and based on its average performance across our objective metrics, we adopt Fx-Encoder++ as the front-end audio encoder for LLM2Fx-Tools.

Table 5: Ablation study on the reverse engineering task using different audio encoders. *Indicates that the metric was computed using the same audio encoder as the input encoder.

| Audio Encoder | Fx-chain | Perceptual Dist. | | DSP | Embedding Sim.($\uparrow$) | | |
|---|---|---|---|---|---|---|---|
| | MAE($\downarrow$) | L/R($\downarrow$) | M/S($\downarrow$) | AF($\downarrow$) | AFx-Rep | FxEnc | FxEnc++ |
| CLAP (Wu* et al., 2023) | 0.21 | 3.84 | 4.21 | 11.3 | 0.55 | 0.52 | 0.53 |
| Fx-Encoder (Koo et al., 2023) | **0.20** | **3.66** | **4.06** | 9.68 | 0.56 | **0.68*** | 0.56 |
| AFx-Rep (Steinmetz et al., 2024) | 0.21 | 3.84 | 3.90 | 10.4 | **0.64*** | 0.58 | 0.52 |
| Fx-Encoder++ (Yeh et al., 2025) | **0.20** | 3.81 | 4.12 | **9.20** | 0.62 | 0.64 | **0.65*** |

## E    EVALUATION METRIC

**Audio Effects Module Classification Accuracy:** We evaluate the model's ability to correctly identify which audio effects module should be applied using standard classification accuracy:

$$\text{Acc} = \frac{\text{correct predictions}}{\text{total predictions}}. \tag{7}$$

**Fx-Chain Order Correlation:** We assess Fx-chain ordering (order of audio effects module) capability using Spearman rank correlation $\rho$ between predicted and ground truth orders. Missing values are set to $|\text{fx\_pool}| + 1$ for consistent ranking evaluation.

**Audio Effects Parameter MAE:** We calculate Mean Absolute Error for parameter prediction: $\text{MAE} = \frac{1}{n}\sum_{i=1}^{n}|\hat{p}_i - p_i|$, where parameters are normalized to $[0, 1]$ before computation. This metric only considers correctly classified effects.

**Left/Right MRS:** Multi-Resolution STFT distance computed separately for stereo channels: $\text{MRS} = \sum_k (\mathcal{L}_{\text{sc}}^{(k)}(\hat{x}_{\text{ref}}, x_{\text{ref}}) + \mathcal{L}_{\text{mag}}^{(k)}(\hat{x}_{\text{ref}}, x_{\text{ref}}))$ , where $k$ indexes different time-frequency resolutions, and

$$\mathcal{L}_{\text{sc}}^{(k)}(x, \hat{x}) = \frac{\left\| |\text{STFT}^{(k)}(x)| - |\text{STFT}^{(k)}(\hat{x})| \right\|_F}{\left\| |\text{STFT}^{(k)}(x)| \right\|_F}, \tag{8}$$

$$\mathcal{L}_{\text{mag}}^{(k)}(x, \hat{x}) = \frac{1}{N} \left\| \log |\text{STFT}^{(k)}(x)| - \log |\text{STFT}^{(k)}(\hat{x})| \right\|_1. \tag{9}$$

**Mid/Side MRS:** We convert stereo audio to Mid/Side representation and compute MRS distance. Mid-channel captures mono content (addition of left and right channels) while Side-channel captures stereo width and spatial characteristics (subtraction of left and right channels).

**DSP Feature Distance:** We extract digital signal processing (DSP) based low-level descriptors, including the root mean square and crest factor, stereo width and stereo imbalance and bark spectrum corresponding to the dynamics, spatialization and spectral attributes respectively.

**Embedding Similarity:** We use pretrained audio effect encoders to extract semantic representations and compute cosine similarity for different types of audio encoders, including AFX-Reps (Steinmetz et al., 2024), and Fx-Encoder (Koo et al., 2023).

## F   PROMPT DETAILS

We present the detailed prompts used for our dataset generation and LLM-as-a-judge evaluation.

**Instruction-Following Generation Prompts:** We use two main prompts for generating our dataset. The first prompt guides the model to generate realistic user-assistant conversations with appropriate tool calls.

```
You are a post-production assistant (mixing and mastering) specialized in
    audio processing and VST plugin chains.
Complete the following conversation.

Output:
[
    {{
        "role": "user",
        "content": [user_instruction]
    }},
    {{
        "role": "assistant",
        "content": [assistant_response]
    }}
]

Tools:
{fx_chain}

Requirements:
- User requests audio effect parameters of the reference audio. {
    str_user_instruction} {str_user_request_specific_fx}
- The reference audio contains {genre} {instrument} sounds.
- In the assistant message, please keep tool number {tool_numer} and the
    tool order {tool_order}
- In the assistant message, briefly explain the audio effect type, order
    and parameters with natural language description. Please provide
    objective information, don't use overly subjective words. Please
    answer with a short and concise description.
```

**Chain-of-Thought Generation Prompts:** The second prompt specifically focuses on generating chain-of-thought reasoning that bridges multimodal understanding with parameter prediction.

```
You are a post-production assistant (mixing and mastering) specialized in
    audio processing and VST plugin chains.
Given a Audio Effects Chain and a previous tool-based chat conversation,
    generate the next chain-of-thought plan.
Return ONLY a single valid JSON object. Do not include any text before or
    after the JSON. Do not use markdown fences.

Outputs:
{{
    "chain_of_thought": "<think>For [task description], Step1,.. Step2,..
        </think>"
}}

Where:
- task description: The task description is the user's request.
```

```
- chain_of_thought: A step-by-step explanation that covers:
- Step 1. From the reference audio, identify the category and order of
    audio effects in the chain. Do not specify exact values.
- Step 2. Create an FX parameter prediction plan that describes the
    general direction and approach for each effect's parameters without
    specifying exact values.

Constraints:
- Use the provided Audio Effects Chain for effect and parameter names;
    match names exactly.
- Chain of thought reflects the assistant's thinking process for analysis
     and parameter prediction.

Audio Effects Chain:
{vst_info}

conversations:
{conversation}
```

**LLM-as-a-Judge Prompts1:** For evaluate dataset generation, we evaluate for tool alignment and thought quality.

```
You are an expert evaluator for audio post-production conversations
    involving VST plugin chains.
Evaluate the assistant's response in the given conversation based on the
    following criteria.

Use scores to show the quality of the response. Here is the detailed
    scoring rubric for evaluating the quality of responses
from AI assistants:
# Tool Alignment (Order, Direction, Parameter Accuracy):
Poor (1): Significant misalignment with tool chain order, incorrect
    parameter directions, and highly inaccurate parameter values that
    would produce undesirable audio results.
Fair (2): Partial alignment with tool order but contains noticeable
    errors in parameter direction or accuracy.
Good (3): Strong alignment with tool order, correct parameter directions,
     and accurate parameter values with only minor room for improvement.
Excellent (4): Perfect alignment with tool chain order, correct parameter
     directions, and highly accurate parameter values demonstrating
    expert-level understanding.

# Thought Quality:
Poor (1): Illogical chain of thought lacking coherent reasoning about
    audio processing decisions.
Fair (2): Basic reasoning but contains gaps in logic or limited
    understanding of audio processing principles.
Good (3): Strong reasoning with clear understanding of effect
    interactions and good audio processing knowledge.
Excellent (4): Expert-level reasoning with sophisticated understanding of
     complex effect interactions.
{{
    "tool_alignment": {{
        "score": [1, 2, 3, 4],
    }},
    "thought_quality": {{
        "score": [1, 2, 3, 4],
    }},
}}

Tool calling ground truth:
{fx_chain}

Conversation to evaluate:
{conversation}
```

**LLM-as-a-Judge Prompts2:** For natual langauge generation, we evaluate for instruction following and chain of thought quality.

```
You are an expert evaluator for audio post-production conversations
    involving VST plugin chains.
Evaluate the assistant's response in the given conversation based on the
    following criteria.

Use scores to show the quality of the response. Here is the detailed
    scoring rubric for evaluating the quality of responses
from AI assistants:
# Instruction Following Quality:
Poor (1): The response does not follow the user's instructions, ignores
    key requirements, or provides irrelevant information. The answer is
    not in natural language or does not address the task described in the
     instruction.
Fair (2): The response partially follows the instructions, but misses
    important details or only addresses some aspects of the user's
    request. The natural language answer may be incomplete or only
    loosely related to the instruction.
Good (3): The response follows the instructions well, addresses most
    requirements, and provides a mostly complete and relevant answer in
    natural language that matches the task in the instruction, but may
    lack some detail or completeness.
Excellent (4): The response fully follows the user's instructions,
    addresses all requirements in detail, and provides a clear, relevant,
     and comprehensive answer in natural language that is directly
    aligned with the task described in the instruction.

# Chain of Thought Quality:
Poor (1): The chain of thought does not logically connect the user's
    query to the assistant's response, lacking coherent reasoning about
    audio processing decisions. The reasoning fails to demonstrate proper
     task decomposition, analysis of user input, and planning for effect
    chain implementation. Or the chain of thought is empty.
Fair (2): The reasoning attempts to bridge the user's query and the
    assistant's response but contains gaps in logic or shows limited
    understanding of audio processing principles. Some evidence of task
    decomposition and planning to handle user input may be present but
    incomplete or flawed.
Good (3): The chain of thought clearly links the user's query to the
    assistant's response, demonstrating effective task decomposition and
    planning. The reasoning provides clear evidence of user input
    analysis and systematic planning to handle requirements with mostly
    sound logic.
Excellent (4): The reasoning expertly bridges the user's query and the
    assistant's response through comprehensive task decomposition and
    strategic planning. The analysis demonstrates thorough task
    decomposition, comprehensive planning to handle user input, and
    expert-level reasoning throughout the process.
{{
    "instruction_following_quality": {{
        "score": [1, 2, 3, 4],
    }},
    "chain_of_thought_quality": {{
        "score": [1, 2, 3, 4],
    }},
}}
Conversation to evaluate:
{conversation}
Chain of thought:
{cot}
```

# G PARAMETER RANGE FOR DATASET SAMPLING

Table 6: Parameter space of the audio effects used in this study. For each parameter, we define the range and discretized step size for both coarse and fine-grained search spaces.

| Parameter | Coarse Range | Step | Fine Range | Step |
|---|---|---|---|---|
| Three-band Equalizer | | | | |
| low gain db | [-20.0, 20.0] | 2 | [-6.0, 6.0] | 1 |
| low cutoff freq | [0.0, 400.0] | 20 | [60.0, 120.0] | 10 |
| low Q factor | [0.0, 6.0] | 0.5 | [0.5, 3.0] | 0.25 |
| mid gain db | [-20.0, 20.0] | 2 | [-6.0, 6.0] | 1 |
| mid cutoff freq | [250.0, 6000.0] | 250 | [250.0, 1000.0] | 100 |
| mid Q factor | [0.1, 6.0] | 0.5 | [0.5, 3.0] | 0.25 |
| high gain db | [-20.0, 20.0] | 2 | [-6.0, 6.0] | 1 |
| high cutoff freq | [4000.0, 20000.0] | 1000 | [4000.0, 8000.0] | 500 |
| high Q factor | [0.0, 6.0] | 0.5 | [0.5, 3.0] | 0.5 |
| Compressor | | | | |
| threshold db | [-40.0, -5.0] | 5 | [-20.0, -10.0] | 1 |
| ratio | [0.0, 20.0] | 1 | [2.0, 8.0] | 0.5 |
| attack ms | [0.0, 500.0] | 5 | [1.0, 30.0] | 1 |
| release ms | [0.0, 1000.0] | 50 | [0.0, 500.0] | 25 |
| Stereo Widener | | | | |
| width | [0.0, 1.5] | 0.1 | [1.1, 1.5] | 0.1 |
| Gain | | | | |
| gain db | [-20.0, 20.0] | 2 | [-6.0, 6.0] | 1 |
| Panner | | | | |
| pan | [-1.0, 1.0] | 0.1 | [-0.6, 0.6] | 0.1 |
| Distortion | | | | |
| drive db | [0.0, 20.0] | 2 | [1.0, 5.0] | 0.5 |
| Reverb | | | | |
| room size | [0.0, 0.9] | 0.1 | [0.3, 0.6] | 0.05 |
| damping | [0.0, 0.9] | 0.1 | [0.3, 0.6] | 0.05 |
| width | [0.0, 0.9] | 0.1 | [0.3, 0.6] | 0.05 |
| mix ratio | [0.0, 1.0] | 0.1 | [0.1, 1.0] | 0.1 |
| Delay | | | | |
| delay seconds | [0.0, 0.7] | 0.05 | [0.01, 0.2] | 0.02 |
| feedback | [0.0, 0.6] | 0.05 | [0.01, 0.2] | 0.02 |
| mix ratio | [0.0, 1.0] | 0.1 | [0.1, 1.0] | 0.1 |
| Limiter | | | | |
| threshold db | [-20.0, -1.0] | 1 | [-5.0, -1.0] | 0.1 |
| release ms | [0.0, 1000.0] | 50 | [0.0, 300.0] | 25 |

