# OpenReview forum: "LLM2Fx-Tools: Tool Calling for Music Post-Production"
_ICLR.cc/2026/Conference — ICLR 2026 Poster_

### Official Review · Reviewer_Cr7x · 2025-10-28

**Soundness:** 3
**Presentation:** 3
**Contribution:** 2
**Rating:** 4
**Confidence:** 4

**Summary:**

This paper proposes a new framework (including a model, dataset, and overall methodology) for music post-production based on a multimodal LLM. This is evaluated on inferring an Fx-chain, but also on style transfer and using the LLM as a judge.

**Strengths:**

* This is a relatively unexplored application domain in terms of using multimodal LLMs for music post-production tasks.
* The overall choice of models, the task definition, training process, dataset creation, and evaluation methodology are all appropriate and technically sound.

**Weaknesses:**

* My main source of criticism for this paper is that this work overall uses established AI methods for a new application. There is little to no AI innovation taking place, and I feel that this work would be more suitable for a venue specializing in audio production or audio engineering (e.g. AES conferences or conventions, ICASSP, or DAFx). I do not see any compelling evidence for inclusion in ICLR.

**Questions:**

As stated above, I fully agree with the design choices made by the authors in terms of methodology, problem setting, evaluation, and the new dataset based on MedleyDB. The paper is also well structured and well written, and I also appreciate the inclusion of a section on Limitations, which is not something always present in ICLR submissions.

My only comment is the one stated above, on whether is ICLR the most suitable venue for a work which does not offer any innovation in AI, but rather uses established AI methods with some minor modifications as to support a research question and problem directly situated in the field of audio engineering. As such I might recommend this paper on being marginally out of scope of ICLR.

---

> ### Author Response · Authors · 2025-11-21
>
> We sincerely thank the reviewer for the thoughtful and constructive feedback. We greatly appreciate your recognition of our work's contributions, particularly exploring a relatively unexplored application domain and the technical soundness of our approach including model selection, task definition, training process, dataset creation, and evaluation methodology. We address the concerns and questions below.
>
> We would like to thank the reviewer for the valuable comment. Here, we address the reviewer's main concern as below.
>
> ```
> My main source of criticism for this paper is that this work overall uses established AI methods for a new application. There is little to no AI innovation taking place, and I feel that this work would be more suitable for a venue specializing in audio production or audio engineering (e.g. AES conferences or conventions, ICASSP, or DAFx). I do not see any compelling evidence for inclusion in ICLR.
> ```
>
> We would like to clarify that ICLR explicitly welcomes applications in the audio domain, according to the Call for Papers (https://iclr.cc/Conferences/2026/CallForPapers):
>
> ```
> "We consider a broad range of subject areas including... applications in vision, audio, speech, language, music... and others."
> ```
>
> **Our work is a strong fit for ICLR. Our work represents a next step in multimodal tool calling** :
>
> Rather than simply selecting which tool to use from a user's text instruction, the LLM must understand the relationship between audio signals and tools. Unlike prior unimodal, text-only tool-calling research, our task requires an LLM to jointly interpret text instructions and two audio inputs, perform difference-aware reasoning, and generate a sequence of tool calls rather than a single prediction.
>
> We introduce the first LLM-driven system for music post-production that integrates conversational interfaces, chain-of-thought audio reasoning, and coordinated use of audio, external VST tools, and LLMs. This task-level, domain-level novelty not only opens a new applied research direction but also showcases how large models can perform real-world actions—an emerging theme within ICLR. For these reasons, we believe our work is particularly well aligned with the conference’s interests.

---

> > ### Comment · Reviewer_Cr7x · 2025-11-25
> >
> > Thank you for the detailed response.
> >
> > I should perhaps clarify, that I do not consider this paper out of scope for ICLR, which based on the call does support papers which include applications to audio among other modalities. Rather, I consider this work to be a better fit to a more domain-specific conference given the contributions made in this work, and given the technically appropriate but perhaps relatively straightforward AI methodologies used in this paper.

---

> ### Author Response · Authors · 2025-11-26
>
> Thank you for the clear response.
>
> First of all, thank you for letting us know that our paper is not out of scope for ICLR. We also apologize if our previous message came across as overly strong in tone. We understand the reviewer’s concern regarding the perceived lack of "AI Innovation" and the suggestion that an audio-specific venue might be more suitable. We agree that our work addresses audio engineering problem, and we seriously considered submission to such conferences. However, we chose ICLR because we believe our paper offers contributions that align closely with the conference’s emerging interests.
>
> First, although our task domain falls within audio engineering, the main focus of our paper is on multimodal large language models. For this reason, we believe that ICLR is an appropriate venue for receiving an accurate and well-aligned review of our work. Our work presents a novel application of multimodal LLMs to music post-production, integrating conversational interfaces, chain-of-thought audio reasoning, and external VST tools. This opens a new applied research direction, showcasing how large language models can perform real-world actions—an emerging theme within ICLR.
>
> Second, while we leverage established AI architectures, our system is not a simple combination of well-known approaches. Simply following a standard multimodal fine-tuning does not necessarily work, often presenting significant practical difficulties. Number prediction is a good example in our case. As reported in our ablation study (Table 2),  we demonstrate that adapting Number Token Loss (NTL) is critical for performance. We would like to highlight that this is the first adaptation of NTL to multimodal LLM fine-tuning with a scientifically grounded motivation.
>
> Lastly, we argue that a well-designed datasets are the prerequisite for AI innovation. In this paper, we publish LP-Fx, a new instruction-following dataset with structured CoT annotations and tool calls for audio effects module. This establishes a novel foundation for the research community, enabling further AI innovation in interpretable, multimodal audio processing system. This moves beyond simply predicting which tool to use; it proposes the next step of tool calling: understanding how the tools should be used.
>
> We believe that this work, as a multimodal audio LLM, represents an important milestone in audio understanding and tool use within the music domain on the path toward general AGI, and we expect it to stimulate active follow-up research

---

### Official Review · Reviewer_A8mk · 2025-10-30

**Soundness:** 3
**Presentation:** 3
**Contribution:** 3
**Rating:** 6
**Confidence:** 4

**Summary:**

This paper presents LLM2Fx-Tools, a tool-calling framework that generates sequences of audio effects (Fx-chains) for music post-production. The authors also introduce a new dataset, LP-Fx, to support this task. The topic is novel and of clear interest to the audio and music research community. The proposed system builds upon Fx-Encoder++ and fine-tunes Qwen-4B to achieve the goal of automatic Fx-chain generation.

**Strengths:**

- The proposed approach to Fx-chain estimation is novel. The integration of Chain-of-Thought (CoT) reasoning into the training framework is also interesting.

- The problem is clearly defined and well motivated.

- The methodology for dataset creation is clearly described and systematically organized.

**Weaknesses:**

- In Figure 1, the meaning of FxNorm is unclear.

- In Figure 2, why does e_{SEP} consist of two tokens?

- Below Equation (1), what is N? What is param_n?

- In Section 2.1, second paragraph, the authors mention “handle both tasks.” What exactly are the two tasks?

- In Section 2.1, the term “secondary task” is introduced but not clearly defined.

- In Section 2.2 (Audio Encoder), why was Fx-Encoder++ chosen over other possible encoders? How might different audio encoders influence system performance?

- The writing in Section 2.3 (Number Token Loss) and Equation (4) needs improvement for clarity. The statement “a key problem with Cross Entropy is that it treats all incorrect predictions equally” is vague—please elaborate on how this issue is addressed in your proposed loss. Overall, Subsection 2.3 and Equation (4) are difficult to follow.

**Questions:**

- Will the training dataset and training code be released for reproducibility in future work?

- Will the evaluation dataset and evaluation code also be made publicly available?

---

> ### Author Response · Authors · 2025-11-21
>
> We thank the reviewer for the thoughtful feedback and for recognizing our contributions, including (i) the novel approach to Fx-chain estimation through structured tool calling, (ii) the integration of Chain-of-Thought (CoT) reasoning into the training framework, (iii) the clearly defined and well-motivated problem formulation, and (iv) the systematic methodology for dataset creation. We address the concerns and questions below.
>
> ### 1. Audio Encoder Ablation Study (Appendix D)
> Thanks for great comments. We conducted an ablation study on the reverse engineering task following the regression approach. We observed that audio effects-specific encoders (Fx-Encoder, AFx-Rep, and Fx-Encoder++) outperform general-purpose audio representations (CLAP). Based on its average performance across our objective metrics, we adopt Fx-Encoder++ as the front-end audio encoder of our LLM2Fx-Tools.
>
> | Audio Encoder | Parameter MAE(↓) | Perceptual Dist. L/R(↓) | Perceptual Dist. M/S(↓) | Perceptual Dist. AF(↓) | Embedding Sim. AFx-Rep(↑) |  Embedding Sim. FxEnc(↑) | Embedding Sim. FxEnc++(↑) |
> |---------------|-----------------|-------------------------|-------------------------|------------------------|-------------------------------|------------------------------|--------------------------------|
> | CLAP (Wu* et al., 2023) | 0.21 | 3.84 | 4.21 | 11.3 | 0.55 | 0.52 | 0.53 |
> | Fx-Encoder (Koo et al., 2023) | 0.20 | 3.66 | 4.06 | 9.68 | 0.56 | 0.68* | 0.56 |
> | AFx-Rep (Steinmetz et al., 2024) | 0.21 | 3.84 | 3.90 | 10.4 | 0.64* | 0.58 | 0.52 |
> | Fx-Encoder++ (Yeh et al., 2025) | 0.20 | 3.81 | 4.12 | 9.20 | 0.62 | 0.64 | 0.65* |
> *Indicates that the metric was computed using the same audio encoder as the input encoder.
>
> - [1] CLAP (Wu* et al., 2023)  Large-scale contrastive language-audio pretraining with feature fusion and keyword-to-caption augmentation
> - [2] Fx-Encoder (Koo et al., 2023), Music mixing style transfer: A contrastive learning approach to disentangle audio
> effects.
> - [3] AFx-Rep (Steinmetz et al., 2024), ST-ITO: Controlling audio effects for style transfer with inference-
> time optimization
> - [4] Fx-Encoder++ (Yeh et al., 2025), Fx-Encoder++: Extracting instrument-wise audio effects representations from mixtures
>
> ### 2. Presentation Clarifications
>
> We appreciate the reviewer's detailed feedback on presentation clarity. We have carefully revised the manuscript to address all the points raised. Below is a summary of the improvements made:
>
> | **Line** | **Reviewer Comment** | **Our Response** |
> |----------|----------------------|------------------|
> | Figure 1 | the meaning of FxNorm is unclear. | → Improved Figure 1 caption |
> | Equation (1) | what is N? What is param_n? | → Added explanation of N and param_n |
> | Section 2.1 | the authors mention "handle both tasks." What exactly are the two tasks? | → Modified to "handle both reverse engineering and blind estimation" |
> | Section 2.1 | the term "secondary task" is introduced but not clearly defined. | → Modified the sentence excluding "secondary task" |
> | Equation (4) |please elaborate on how this issue is addressed in your proposed loss. | → Modified Equation (4)'s notations and explanations on Number Token Loss. |
>
> ### 3. Why does e_{SEP} consist of two tokens?
>
> Thank you for the great question. We agree that this detail was missing in the paper, and we have added it in the revised version. We used the separation tokens not as special tokens, but as natural-language tokens that are tokenized into standard vocabulary indices ('reference_audio': [16291, 7699], 'dry_audio': [53210, 7699]). Our intention was to create token overlap between the user’s input instruction and the audio indicators. Based on our empirical experiments, using natural-language separators yielded slightly better performance than expanding the vocabulary and introducing two additional special tokens.
>
> ### 4. Open Source for Reproducibility
> We plan to open-source resource (dataset, code) for the benefit of open community.

---

### Official Review · Reviewer_CSNR · 2025-11-01

**Soundness:** 2
**Presentation:** 4
**Contribution:** 3
**Rating:** 8
**Confidence:** 3

**Summary:**

The paper presents LLM2Fx-Tools, a novel tool-calling framework which for a given set of audio inputs, provides executable audio effects sequences (Fx-chain), with appropriate CoT reasoning and responses . The paper also introduces LP-FX, a new instruction following dataset with CoT annotations and tools calls for audio effects. The authors provide experimental validation for their approach along with a demo page for subjective verification.

**Strengths:**

Originality: The paper's key novelty lies in formulating Fx-chain estimation as a LLM-based tool call problem. The autoregressive modeling for LLMs is able to learn the sequential order of audio effect calls as opposed to systems only based on audio features.

Quality: The paper has detailed experiments around the three evaluation tasks, reverse engineering to show the model can predict tool-chain for paired audios, blind style transfer to show the generalization capability to unseen audios, and natural language language generation to showcase interpretability. Across all the tasks,  LLM2FX-Tools results are strong as compared to the baselines. The authors conduct ablations to show the importance of optimization decisions (CoT, NTL, MST).

Clarity: The paper is well written with clear notation and figures, with appendix covering all necessary details for dataset generation, evaluation and LLM prompting.

Significance: LLM2FX-Tools framework treats the audio effect modules as external non-differentiable tools, which makes the framework flexible to diverse real world scenarios.The authors also present a LP-FX dataset with CoT annotations and tool calls, which is beneficial for future research

**Weaknesses:**

For the reverse engineering task, the strongest baseline is Multi-task regression, which comes close even without relying on the ordering of Fx-chain, while the LLM is learning that information. The authors can consider adding a pairwise-ordering loss for the 9 audio effects for the multi-task baselines.

For the style transfer task, the style of the output appears to be mixed between the input and reference audio while listening subjectively to the demo examples. A comparison with differential audio effects style transfer baseline would be quite important to see (https://arxiv.org/pdf/2207.08759). The objective evaluation can benefit on a larger set than 100 test samples

As covered in the limitation section, the paper relies on Fx-normalization and Fx-removal preprocessing, while ideally, they should be modeling as part of the tool-calling framework. Experimental validation is limited to single instruments, datasets are relatively smaller in size with ~2k tracks.

**Questions:**

For equation 4, $N$ corresponds to the sequence length or training examples? Previous equation 3 has $t$, which is over the sequence, while for $t$ in equation 4 represents the upper range of the number token. It will be helpful if we can improve the notation a little here.

Please consider having a subjective evaluation and a stronger baseline like (https://arxiv.org/pdf/2207.08759) for the style transfer task.

Gemini 2.5 Pro is used both for dataset generation judge in 3.2, and natural language evaluation judge for table 4. Could we use a different judge to remove the bias for this case?

---

> ### Author Response · Authors · 2025-11-21
>
> We thank the reviewer for their thoughtful comments and for recognizing the strengths of our work: (1) Originality—formulating Fx-chain estimation as an LLM-based tool-calling problem; (2) Quality—comprehensive experiments with results and ablations; (3) Clarity—clear presentation with detailed appendix; and (4) Significance—a flexible framework and the LP-FX dataset for future research. We address the concerns and questions below.
>
> ### 1. Differentiable audio effects style transfer as a baseline
>
> [line 310, Table 2, Table 3] We thank the reviewer for this insightful comment and strongly agree. We have added a differentiable audio effects baseline [1] (DeepAFx-ST) using differentiable audio effects modules from [2] trained with perceptual loss (MR-STFT) as another regression baseline. We report results for both reverse engineering and style transfer tasks. DeepAFx-ST demonstrates strong performance, but lacks the key advantages of LLM2Fx: (1) tool planning, (2) interpretability through chain-of-thought reasoning, and (3) response generation capabilities. We believe this baseline clearly highlights both the limitations and strengths of LLM2Fx, and we have included it in the updated version.
>
> | Method | Acc. (↑) | Corr. (↑) | MAE (↓) | MR STFT L/R (↓) | MR STFT M/S (↓) | AF (↓) | AFx-Rep Sim (↑) | FxEnc Sim (↑) |
> |--------|-------------------|--|--|------------------|--|-----|----------------|--|
> | No Fx | - | - | - | 13.11 | 13.49 | 14.82 | 0.50 | 0.30 |
> | Random Fx | 52% | -0.01 | 0.39 | 8.07 | 8.90 | 13.70 | 0.41 | 0.34 |
> | Regression | 55% | -0.03 | 0.20 | 3.81 | 4.12 | 9.20 | 0.62 | 0.64 |
> | MultiTask | 61% | 0.00 | 0.23 | 3.17 | 3.39 | 8.39 | 0.63 | 0.66 |
> | DeepAFx-ST* | - | - | - | 1.75 | 2.06 | 3.95 | 0.62 | 0.66 |
> | Gemini2.5-Flash | 78% | 0.54 | 0.32 | 3.42 | 4.24 | 14.97 | 0.56 | 0.50 |
> | LLM2Fx-Tools | 80% | 0.56 | 0.23 | 3.13 | 3.27 | 8.29 | 0.68 | 0.67 |
>
> *DeepAFx-ST was trained with Perceptual Dist. as its training objective.
>
> - [1] DeepAF-ST, Style Transfer of Audio Effects with Differentiable Signal Processing, Journal of the Audio Engineering Society
> - [2] dasp-pytorch, https://github.com/csteinmetz1/dasp-pytorch
>
> ### 2. LLM-as-a-judge bias
>
> [Table 4] We appreciate this valuable observation and strongly agree with the concern about potential bias. To address this, we have replaced the Gemini 2.5 Pro judge (which was used for dataset generation and filtering in Section 3.2) with GPT-5 as the evaluation judge for Table 4. This change eliminates any potential bias from using the same model for both data generation and evaluation, ensuring a more independent and fair assessment of our method's performance. We have updated Table 4 (Natural Language Generation Results) by replacing Gemini 2.5 Pro to GPT-5 as a judge to demonstrate consistency across different evaluators. Comparison between these two models is as follow:
>
> | Model | Gemini 2.5 Pro IF Quality | Gemini 2.5 Pro CoT Quality | GPT-5 IF Quality | GPT-5 CoT Quality |
> |-------|---------------------------|----------------------------|-------------------|-------------------|
> | Qwen 2.5 Omni | 2.01 | N/A | 1.46 | N/A |
> | Qwen 3 | 3.63 | 3.62 | 2.89 | 2.30 |
> | Gemini 2.5 Flash | 3.75 | 3.63 | 3.39 | 3.03 |
> | **LLM2Fx-Tools** | **3.86** | **3.69** | **3.50** | **3.05** |
>
> The results show that LLM2Fx-Tools consistently achieves the highest scores across both judges, demonstrating the robustness of our approach. While absolute scores differ between judges (as expected due to different evaluation criteria), the relative ranking remains consistent, validating our method's superiority.
>
> ### 3. Subjective evaluation
>
> We appreciate the reviewer's suggestion for subjective evaluation. We are now conducting a listening test evaluation. We will share the results once the study is complete.
>
> ### 4. Presentation Clarifications
>
> [Equation 4, Section 5] We thank the reviewer for pointing out the presentation issues. We have rewritten Equation 4 to provide clearer mathematical notation and better explain the number token loss. Also, we updated the limitation section addressing the small size of the test set.

---

> > ### Comment · Reviewer_CSNR · 2025-11-27
> >
> > Thank you for adding the DeepAFx-ST baseline, it highlights gap in MR STFT metrics which LLM based Fx-chain estimation methods can attempt to close as a future work. The novelty of Fx-chain estimation as a LLM-based tool call problem still remains a key strength of the paper.
> >
> > I also thank the authors for updating GPT-5 as the LLM judge to remove the bias, and look forward to seeing subjective evaluation results.

---

> > > ### Author Response · Authors · 2025-12-01
> > >
> > > We appreciate the reviewer's suggestion to conduct a subjective evaluation. To evaluate the perceptual quality of reverse engineering results, we conducted a MUSHRA (MUltiple Stimuli with Hidden Reference and Anchor) listening test. Participants were asked to rate different mixes based on their similarity to the reference mix in terms of audio effects characteristics. Each test question consisted of a reference track and seven different stimuli tracks. We designed the test with a total of 8 questions and 16 participants.
> > >
> > > The table below summarizes the MUSHRA test results:
> > >
> > > | System                 | MUSHRA Score (Mean ± Std) |
> > > |------------------------|----------------------------|
> > > | Reference              | 81.50 ± 20.13              |
> > > | LLM2Fx-Tools (Ours)    | 62.75 ± 22.27              |
> > > | Gemini 2.5 Flash       | 56.51 ± 25.92              |
> > > | DeepAFX-ST             | 54.77 ± 24.19              |
> > > | No Fx                  | 39.10 ± 24.74              |
> > > | MultiTask              | 34.92 ± 27.70              |
> > > | Regression             | 16.21 ± 12.78              |
> > >
> > >
> > > The hidden reference achieves a mean score of 81.5, confirming the validity of our test design. Our proposed LLM2Fx-Tools outperforms other methods. Pairwise t-tests reveal that LLM2Fx-Tools (62.8), Gemini 2.5 Flash (56.5), and DeepAFX-ST (54.8) all significantly outperform the No Fx baseline (39.1) with p<0.05.
> > >
> > > While Gemini 2.5 Flash and DeepAFX-ST show no statistically significant difference between each other (p=0.747), LLM2Fx-Tools significantly outperforms both Gemini 2.5 Flash (p=0.020) and DeepAFX-ST (p=0.006). Interestingly, both the MultiTask (34.9) and Regression (16.2) baselines score lower than the No Fx baseline, indicating that incorrect effect application can degrade perceptual quality more than applying no effects at all.
> > >
> > > This finding reinforces our earlier observation that effective Fx-chain planning (including proper effect selection and ordering) is important for achieving human-level similarity.

---

### Official Review · Reviewer_Rbd9 · 2025-11-04

**Soundness:** 3
**Presentation:** 3
**Contribution:** 2
**Rating:** 4
**Confidence:** 3

**Summary:**

This paper applies existing LLM tool calling techniques to audio effects chain generation. The system uses chain-of-thought to predict effect sequences from audio. The authors create a 101K synthetic dataset LP-Fx generated by Gemini 2.5. In my opinion, the work is mostly an application of existing techniques to a new domain without significant technical innovation.

**Strengths:**

1. First work applying structured tool calling to audio effects chains
2. Comprehensive evaluation across multiple metrics

**Weaknesses:**

1. The paper misuses terminology. "Audio style transfer" has established meaning in audio processing literature (timbre/texture transformation). This work only does audio effects parameter transfer, which is much narrower. This creates confusion with existing work and is misleading.
2. Limited technical novelty. The method is standard multimodal LLM fine-tuning: audio encoder -> adapter -> LLM with LoRA. This is direct application of existing techniques without methodological contribution.
3. No human evaluation despite claims about "interpretability" and "controllable music production". All evaluation is automatic metrics or LLM-as-a-judge, which has known reliability issues.
4. Missing details: How does the model handle effects outside the 9 trained types? The paper claims "users can incorporate their own audio plugins" but provides no evidence.
5. The work is mostly experimental validation that LLM tool calling works for this task. The technical contribution is limited.

**Questions:**

Recommand using the term "audio effects (parameter) transfer" instead of "audio style transfer"

Gemini 2.5 Flash gets MAE 0.32 vs your 0.23, but it achieves 78% accuracy vs your 80%. Why does such a large model fail so badly on parameters? Does it indicate setup issue?

---

> ### Author Response · Authors · 2025-11-21
>
> We thank the reviewer for the thoughtful feedback and for recognizing our contributions, including (i) being, to our knowledge, the first to apply structured tool calling to audio effects chains, and (ii) providing a comprehensive evaluation across multiple metrics. We address the concerns and questions below.
>
>
> ### 1.Novelty of LLM2Fx-Tools
>
> Our work represents a next step in multi-modal tool calling: rather than simply selecting which tool to use from a user's text instruction, our LLM2Fx-Tools is designed to understand the relationship between audio signals and tools.
>
> Prior LLM tool-calling works [1,2] operate in a unimodal behavior, selecting tools solely from text. Although recent works have explored tool calling for audio processing [3,4], these approaches neither truly understand the audio input nor predict any internal tool parameters, and remain constrained to tool classification only. In contrast, our model directly listens and analyzes audio signals to predict a sequence of tool modules (audio effects chain), predicting both (a) the order of tools and (b) the corresponding numerical parameters.
>
> Our work's novelty lies in the task formulation: a multimodal tool-calling framework that executes tool calls based on a text instruction and two audio inputs, and is difference-aware: by listening to both audio files, the model explicitly reasons how the processed audio differs from the reference and uses that acoustic understanding to guide tool selection and parameterization.
>
> * [1] Toolformer: Language models can teach themselves to use tools. Advances in Neural Information Processing Systems, 2023
> * [2] ToolAlpaca: Generalized Tool Learning for Language Models with 3000 Simulated Cases, ArXiv, 2023
> * [3] AudioAgent: Enhancing Task Performance Through Modality-Driven Prompt Optimization, ArXiv, 2024
> * [4] Audio-Agent: Leveraging LLMs For Audio Generation, Editing and Composition, ArXiv, 2024
>
> ### 2. Gemini's performance on accuracy and MAE
>
> Gemini 2.5 Flash demonstrates a solid general understanding of which audio effect types are applied to the input audio. However, it lacks precise understanding of post-production effects and their continuous parameter values, which might be an indication that such large general purpose models may have not been trained on highly specialized audio engineering tasks (zeroshot task for Gemini 2.5 Flash).
>
> In contrast, our model is explicitly trained to predict the effect parameters as part of the target sequence, and the Number Token Loss (NTL) directly improves numerical accuracy. Notably, when we remove NTL from our model, the MAE increases to 0.32, which is the same performance as Gemini's. This suggests that precise numerical prediction is crucial for such regression task, and that domain-aligned training objectives make a significant difference.

---

> ### Author Response · Authors · 2025-11-21
>
> ### 3. Claims about "interpretability" and "controllability"
>
> We thank the reviewer for raising this important point. Our claims about controllability and interpretability stem from the natural language interface that bridges user intent and effect parameters.
>
> **Controllability:** Regression-based approaches predict parameters solely from audio input, lacking natural language understanding. In contrast, LLM2Fx-Tools accepts natural language instructions alongside audio, enabling users to query specific aspects of the audio processing (e.g., "I'm looking for a reverb to achieve a similar sound.?"). This instruction-following capability provides fine-grained control over what information the model extracts, which is directly reflected in our Instruction Following (IF) quality metrics.
>
> **Interpretability:** While regression models directly output parameters, LLM2Fx-Tools generates chain-of-thought reasoning that explains the prediction process in natural language. This allows users to understand *why* certain parameters were predicted, making the model's decision-making process transparent. This interpretability is captured by our Chain-of-Thought (CoT) quality metrics.
>
> Our work proposes this novel framework where interpretability and controllability are directly measured through LLM-as-a-judge evaluations using both CoT and IF prompts, as shown in the multi-judge results above.
> We appreciate the concern about LLM-as-a-judge reliability. To ensure robust evaluation, we employ multiple judge models rather than relying on a single judge. As shown in the table below, we report results using four different judge configurations: Gemini 2.5 Pro and GPT-5. This multi-judge approach provides a more comprehensive and reliable evaluation.
>
> | Model | Gemini 2.5 Pro IF Quality | Gemini 2.5 Pro CoT Quality | GPT-5 IF Quality | GPT-5 CoT Quality |
> |-------|---------------------------|----------------------------|-------------------|-------------------|
> | Qwen 2.5 Omni | 2.01 | N/A | 1.46 | N/A |
> | Qwen 3 | 3.63 | 3.62 | 2.89 | 2.30 |
> | Gemini 2.5 Flash | 3.75 | 3.63 | 3.39 | 3.03 |
> | **LLM2Fx-Tools** | **3.86** | **3.69** | **3.50** | **3.05** |
>
>
> ### 4. Terminology "Audio style transfer"
>
> Thank you for the suggestion. We noticed that the phrase “audio style transfer” appeared once in the conclusion, and we have updated that for consistency.  Currently, throughout the paper the task is always referred to as “Audio Effects Style Transfer”.
>
> ### 5. How does the model handle effects outside the 9 trained types?
>
> Thank you for the great feedback. We confirm that our paper does not include experiments on zero-shot (unseen) audio effects. We noticed that the abstract contained the sentence “users can incorporate their own audio plugins,” and we confirmed that this was an overclaim, so we have removed it. We also added this point to the limitations section to clarify that it can be addressed in future work. (line 490).
>
> We sincerely appreciate the reviewer’s thoughtful and constructive comments. We have made every effort to incorporate the suggestions as fully as possible, and we are grateful for the guidance that helped us improve the quality and clarity of the paper.

---

### Author Response · Authors · 2025-12-01
**Summary of Contributions and Rebuttal Updates for the Area Chair**

Dear Area Chair,

We would like to thank the reviewers for their thoughtful and constructive feedback. During the rebuttal period, we engaged in productive discussions and have successfully addressed the concerns raised by the reviewers through additional experiments and clarifications.

### **A Novel Framework for Multimodal Tool Use.**
This paper introduces the first framework to treat audio effects chain estimation as an interpretable Tool-Calling problem. By enabling a Multimodal LLM to "listen" to differences between audio signals and generate executable actions via Chain-of-Thought (CoT) reasoning, we bridge the gap between abstract language reasoning and continuous signal processing.

Beyond performance, our framework offers two critical advantages over baseline approaches: **Controllability**—regression-based approaches predict parameters solely from audio input, lacking natural language understanding, whereas LLM2Fx-Tools accepts natural language instructions alongside audio, enabling users to query specific aspects of the audio processing (e.g., "I'm looking for a reverb to achieve a similar sound") and allowing for interactive, user-guided exploration of the effects space; and **Interpretability**—while regression models directly output parameters as black-box predictions, LLM2Fx-Tools generates chain-of-thought reasoning that explains the prediction process in natural language, enabling users to understand **why** certain effects were chosen and **how** the model arrived at specific parameter values, making the system transparent and debuggable.

The reviewers have broadly recognized the originality and rigor of our approach. Reviewer CSNR (Score: 8) championed the work for its "originality in formulating Fx-chain estimation as a tool-call problem" and the strength of the evaluation. Reviewer A8mk (Score: 6) highlighted the novelty of integrating CoT into this framework. Even the reviewers (Rbd9, Cr7x) acknowledged that the methodology is "technically sound," the problem is "well-motivated," and the dataset creation is "systematic." The consensus is that the work is of high novelty and suitable for ICLR.

###  **Resolving Concerns through Rebuttal.**
We have actively resolved the primary concerns raised during the review process. We emphasize that these are not just arguments, but completed experiments and revisions:

- Subjective Human Evaluation [CSNR & Rbd9]. Both reviewers noted the lack of human listening tests. We successfully conducted a MUSHRA listening test (N=16) during the rebuttal. The results demonstrated that our method achieves a statistically significant improvement over baselines in human perceptual quality.

- Stronger Baselines [CSNR]. To address the request for a non-LLM comparison, we implemented and reported results for the DeepAFx-ST (differentiable signal processing) baseline. This confirmed that our LLM-based tool-use approach outperforms differentiable signal processing baselines.

- Audio Encoder Ablation Study [A8mk]. We conducted an ablation study comparing different audio encoders, including audio effects-specific representations and audio-text representations. This study clearly justified our model design choice.

- Evaluation Bias [CSNR, Rbd9]. We addressed the concern of using Gemini as both a generator and judge by conducting evaluations with both Gemini 2.5 Pro and GPT-5 as judges. The consistent trends across both models confirmed the absence of model-specific bias and validated the robustness of our LLM-as-a-judge evaluation methodology.

###  **Alignment with ICLR.**
Finally, we address the "venue fit" comments [Rbd9 & Cr7x] suggesting this work belongs in an audio engineering conference. We argue that ICLR is the ideal home for this research. Our primary contribution is not signal processing theory, but the capabilities of Large Language Models—specifically, how general-purpose models can learn to use tools and reason in audio domains. This work makes significant contributions to two underexplored areas in the LLM research community: (1) **Multimodal Tool-Calling**—while tool use has been extensively studied in text-only settings, research on multimodal tool-calling remains limited, and (2) **Difference Understanding**—the ability of LLMs to comprehend and reason about the differences between two multimodal signals is a largely unexplored capability that is critical for multimodal tasks. Our work demonstrates that LLMs can successfully bridge these gaps, opening new research directions for the ICLR community in multimodal reasoning and tool use.

After rebuttal period, We are confident that the added human evaluations and the demonstrated novelty of the tool-use framework make this a solid contribution to the conference. We hope you will consider these points in your final decision.

Sincerely, The Authors of LLM2Fx-Tools

---

### Meta-Review · Area_Chair_SPgJ · 2026-01-09

**Summary:**

The paper introduces LLM2Fx-Tools, a framework formulating audio effects chain estimation as a multimodal tool-callingproblem. Itleverages an LLM to "lIisten" to audio diferences and generate executable effect sequences with parameters, guidedReChotleef to t CGntraranife The uthorf alfenforfeterfatcory dnetdteerofrgo ife ctareral for thlf fasl abouttechnical novelty and venue fit, the authors successfully demonstrated that LLM-based reasoning provides superiorinterpretability and controllability over black-box regression models.

**Reviewer Concerns:**

Addressed Concerns:
Subjective Human Evaluation (Addressed): Authors conducted a MUSHRA listening test (N=16) showing LLM2Fx-Toolssignificantly outperforms baselines in perceptual quality (p < 0.05).Stronger Baselines (Addressed): Added DeepAFx-ST as a differentiable signal processing baseline; LLM2Fx-Tools achieved
higher human similarity scores.Evaluation Bias (Addressed): Replaced Gemini 2.5 Pro with GPT-5 as a judge for metrics, confirming consistent results and
removing model-specific bias.Audio Encoder Justification (Addressed): Conducted an ablation study justifying the choice of Fx-Encoder++ over generalrepresentations like CLAP.
Outstanding Concerns:Technical Innovation (Remaining): Reviewer Cr7x and Rbd9 still consider the Al methodologies (LoRA fine-tuning, standardadapters) as relatively straightforward applications of existing tech.Style Transfer Ambiguity: While renamed to "Audio Effects Style Transfer," the scope remains focused on parameter transferrather than broader timbre/texture transformation.

**Reviewer Scores:**

Reviewer CSNR (Current: 8): Remains 8. Explicitly satisfied with the DeepAFx-ST baseline and the removal of judge bias.Reviewer A8mk (Current: 6): Likely stays 6 or moves to7. Their presentation concerns and encoder questions were fullyaddressed in Appendix D.Reviewer Rbd9 (Current: 4): Likely moves to 5. Their major "Weakness 3"(No human evaluation) was directly resolved bythe MUSHRA test.Reviewer Cr7x (Current: 4): Likely moves to 5. They conceded that the paper is not out of scope for ICLR but stillfeel anaudio-specific venue is a better fit.

---

### Decision · Program_Chairs · 2026-01-26

Accept (Poster)